# CaReTS: A Multi-Task Framework Unifying Classification and Regression for Time Series Forecasting

## Abstract

Recent advances in deep forecasting models have achieved remarkable performance, yet most approaches still struggle to provide both accurate predictions and interpretable insights into temporal dynamics. This paper proposes CaReTS, a novel multi-task learning framework, that combines classification and regression tasks for multi-step time series forecasting problems. The framework adopts a dual-stream architecture, where a classification branch learns the stepwise trend into the future, while a regression branch estimates the corresponding deviations from the latest observation of the target variable. The dual-stream design provides more interpretable predictions by disentangling macro-level trends from micro-level deviations in target variable. To enable effective learning in output prediction, deviation estimation, and trend classification, we design a multi-task loss with uncertainty-aware weighting to adaptively balance the contribution of each task. Furthermore, four variants (CaReTS1–4) are instantiated under this framework to incorporate the mainstream temporal modelling encoders, including convolutional neural networks (CNNs), long short-term memory networks (LSTMs), and Transformers. Experiments on real-world datasets demonstrate that CaReTS outperforms state-of-the-art (SOTA) algorithms in forecasting accuracy, while achieving higher trend classification performance.

## 1 Introduction

Time series forecasting is a fundamental problem for a wide range of applications, including energy demand management (Grandón et al., 2024), financial data analysis (Bhambu et al., 2024), healthcare monitoring (Ni et al., 2024), and climate modeling (Hittawe et al., 2024). Accurate multi-step forecasting is particularly critical to enable informed decision-making that can capture short- and long-horizon temporal dynamics of the system. Despite its importance, multi-step forecasting remains challenging: prediction accuracy typically decreases as the forecast horizon increases (Yao et al., 2025b), while model interpretability is often limited, reducing trust in high-stakes scenarios (Chakraborty et al., 2024).

The past decade has witnessed remarkable progress through deep learning. Early approaches employed convolutional neural networks (CNNs) to capture local temporal patterns (Wibawa et al., 2022; Durairaj & Mohan, 2022), as well as recurrent neural networks (RNNs) such as long short-term memory (LSTM) and gated recurrent unit (GRU) to model sequential dependencies (Waqas & Humphries, 2024; Yunita et al., 2025). Most recently, Transformers (Vaswani et al., 2017) have emerged as the dominant backbone for both short- and long-horizon forecasting, with many variants improving efficiency and representation: Informer (Zhou et al., 2021) introduced ProbSparse attention; Autoformer (Wu et al., 2021) applied seasonal–trend decomposition with autocorrelation attention; FEDformer (Zhou et al., 2022) leveraged frequency-domain filtering; PatchTST (Nie et al., 2022) utilized patch-based embeddings; and iTransformer (Liu et al., 2023) inverted the modeling axis to focus on variable dependencies. Collectively, these advances significantly improved accuracy on various benchmarks (Wang et al., 2024b). More related work can be found in Appendix A.1.

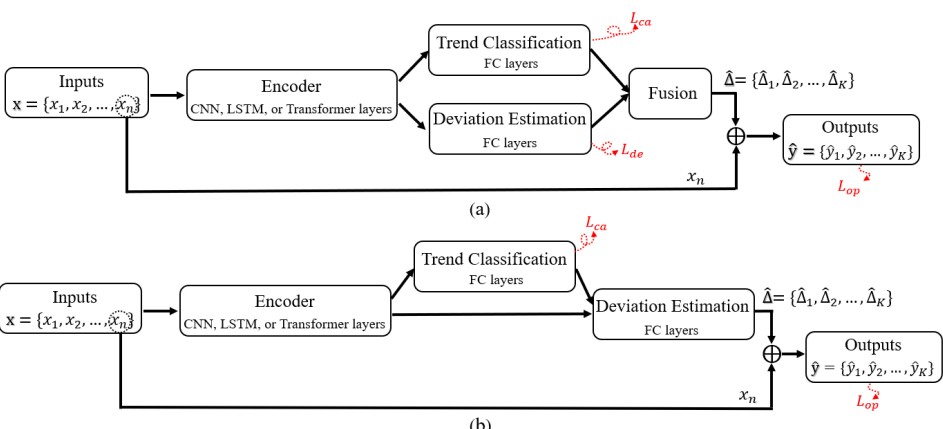

Figure 1: Two types of dual-stream CaReTS architectures

Despite these advances, most deep forecasting models still formulate forecasting as a single regression task, focusing exclusively on numerical prediction of future values. This design makes it difficult to disentangle macro-level future trends (e.g., upward or downward trajectories) from micro-level deviations, thereby limiting interpretability and robustness in multi-step settings (Wang et al., 2018; Tessier & Armstrong, 2015). To overcome these limitations, we propose CaReTS, a multi-task learning framework that unifies classification and regression for multi-step time series forecasting. This framework not only improves prediction accuracy but also enhances interpretability by disentangling macro-level trends from micro-level deviations. Our main contributions are summarized as follows:

- CaReTS introduces a dual-stream architecture, where a classification branch predicts stepwise macro-level trends, and a regression branch estimates fine-grained deviations relative to the latest observation.
- CaReTS designs a multi-task loss with uncertainty-aware weighting to jointly optimize classification and regression tasks, adaptively learning their contributions.
- Four variants (CaReTS1–4) are instantiated to work alongside mainstream temporal encoders (e.g., CNNs, LSTMs, and Transformers), demonstrating the framework's compatibility with diverse modeling paradigms.
- Extensive experiments show that CaReTS achieves state-of-the-art accuracy while providing enhanced interpretability with manageable computational overhead.

## 2 CaReTS Framework

This section introduces a novel multi-task learning framework for multi-step time series forecasting - CaReTS. Specifically, two types of CaReTS architectures are presented, each consisting of a classification branch that captures the stepwise trend of future values and a regression branch that estimates the corresponding deviations. Moreover, a multi-task loss formulation, together with an uncertainty-based loss weighting algorithm, is designed to jointly optimize three tasks including the output prediction, deviation estimation, and trend classification.

### 2.1 CaReTS Architecture

Unlike traditional regression-based approaches that directly predict future values, this work designs two types of dual-stream CaReTS architectures that combine classification and regression tasks, as illustrated in Figure 1. In both architectures, time series models such as CNNs, LSTMs, and Transformers are employed to encode temporal features from the input sequence $\mathbf{x} = \{x_1, x_2, \ldots, x_n\}$. Here, $n$ denotes the total number of input variables, with the last entry $x_n$ must denote the most recent observation of the target variable.

The encoded temporal features are then processed through dual-stream pathways but differ in their fusion strategies. In architecture (a), these features are fed in parallel into two separate fully connected (FC) streams: a classification stream to model the stepwise trend (i.e., upward or downward) and a regression stream to estimate the corresponding deviations relative to the latest observation $x_n$. The final prediction $\hat{\mathbf{y}} = \{\hat{y}_1, \hat{y}_2, \ldots, \hat{y}_K\} = \{\hat{x}_{n+1}, \hat{x}_{n+2}, \ldots, \hat{x}_{n+K}\}$ is obtained in a residual form to fuse the outputs from both streams, i.e. the sum of $x_n$ and the predicted deviation $\hat{\mathbf{\Delta}}$. In contrast, architecture (b) adopts a sequential dual-stream design. The encoded temporal features are first processed by the classification stream to infer the trend. The resulting classification output is then concatenated with the original temporal features and passed into the regression stream for deviation estimation. Therefore, a separate fusion module is no longer required. Finally, similar to architecture (a), the final predictions $\hat{\mathbf{y}}$ are produced by combining the predicted deviations $\hat{\mathbf{\Delta}}$ with the latest observation $x_n$.

## 2.2 MULTI-TASK LEARNING

Building upon the dual-stream architectures introduced above, the CaReTS framework adopts a multi-task learning strategy to jointly model three interrelated tasks: trend classification, deviation estimation, and output prediction. This design is intended to improve forecasting accuracy while enhancing interpretability by explicitly separating the modelling of trend from that of magnitude. In architecture (a), all three tasks are learned in parallel. The overall loss function $L_{(a)}$ is formulated as:

$$L_{(a)} = \alpha_{ca} L_{ca} + \alpha_{de} L_{de} + \alpha_{op} L_{op} \tag{1}$$

where $L_{ca}$ corresponds to the trend classification loss, evaluating the correctness of predicted trend (e.g., upward or downward movement) across multiple future steps; $L_{de}$ represents the deviation estimation loss, responsible for quantifying the magnitude of deviations at each step relative to $x_n$; $L_{op}$ denotes the output prediction loss, aimed at minimizing the discrepancy between the final predicted value $\hat{\mathbf{y}}$ and the ground truth; $\alpha_{ca}, \alpha_{de}$, and $\alpha_{op}$ are the balancing weights of each tasks. A detailed formulation of each loss will be presented in Section 3.

In contrast, architecture (b) simplifies the learning objective to two tasks, as it does not use an explicit fusion module. Here, the deviation estimation and output prediction are effectively combined into a single regression task, resulting in the following loss function:

$$L_{(b)} = \alpha_{ca} L_{ca} + \alpha_{op} L_{op} \tag{2}$$

By structuring the prediction process into distinct but related tasks, the CaReTS framework facilitates more transparent forecasting. It explicitly models how the trend influences the predicted output, thereby offering valuable insights for multi-step time series prediction.

Optimizing the three interrelated tasks simultaneously poses significant challenges, particularly due to discrepancies in loss scales, convergence dynamics, noise levels, and potential conflicts between task objectives. To address these issues, an uncertainty-based loss weighting algorithm (Kendall et al., 2018) is employed to adaptively adjust the contribution of each task during training. As defined in (3), each task's weight is modelled as the inverse of its predicted variance, reflecting the principle that tasks with higher uncertainty should contribute less to the overall loss. As a result, $\alpha_{ca}, \alpha_{de}$, and $\alpha_{op}$ are not treated as static hyperparameters but are instead parameterized through learnable variables that capture the relative confidence of the model in each task. This formulation enables the model to focus on more informative and reliable tasks throughout the optimization process, thereby improving training stability and predictive performance.

$$\alpha_i = \frac{1}{2\sigma_i^2}, \quad i \in \{ca, de, op\} \tag{3}$$

where $\sigma_{ca}^2, \sigma_{de}^2$, and $\sigma_{op}^2$ represent the predicted variance (uncertainty) for each task.

In our implementation, the uncertainty-based task weights are modelled through their logarithmic counterparts $(\log \sigma_i^2)$ to improve numerical stability and allow unconstrained gradient-based optimization. Specifically, each $(\log \sigma_i^2)$ is treated as a learnable parameter, and the corresponding task weight is derived via exponential transformation. Accordingly, the overall loss function for architecture (a) and (b) are reformulated as:

$$L_{(a)} = \sum_{i \in \{\text{ca,de,op}\}} \left( \frac{1}{2} e^{-\log \sigma_i^2} L_i + \frac{1}{2} \log \sigma_i^2 \right) \tag{4}$$

$$L_{(b)} = \sum_{i \in \{\text{ca,op}\}} \left( \frac{1}{2} e^{-\log \sigma_i^2} L_i + \frac{1}{2} \log \sigma_i^2 \right) \tag{5}$$

This formulation integrates both adaptive loss weighting and uncertainty regularization, allowing the model to automatically calibrate the relative importance of each subtask throughout training. As a result, the CaReTS framework achieves improved robustness and predictive performance in multi-step time series forecasting, while preserving interpretability by explicitly modelling task-specific uncertainty.

It should be noted that two stabilization strategies are used here to prevent pathological solutions (e.g., the model assigning arbitrarily large uncertainty to minimize its contribution). On the one hand, each log-variance parameter is softly regularized by an additional penalty term added to the total loss. On the other hand, during training, the log-variance values are constrained within a bounded range $[-10, 10]$ via clamping. These mechanisms jointly help stabilize uncertainty learning and avoid degenerate minima.

## 3 CaReTS Models

This section presents the design of the proposed approaches, i.e., the temporal encoder and the CaReTS models. To ensure broad applicability rather than introducing novel feature extractors, we adopt three mainstream temporal modeling algorithms, CNNs, LSTMs, and Transformers, as interchangeable encoders for extracting sequential features from the input time series. The structures of these encoders are illustrated in Appendix A.2. The primary focus of this section is the design of the CaReTS models. Building upon the dual-stream CaReTS architectures introduced in Section 2.1, four specific model variants (CaReTS1, CaReTS2, CaReTS3, and CaReTS4) are developed to explore different strategies for combining classification-based trend modelling with regression-based deviation estimation, as illustrated in Table 1. Specifically, CaReTS1–CaReTS3 adopt architecture (a), where the two streams operate in parallel, and CaReTS4 adopts architecture (b), where the trend prediction precedes and conditions the deviation estimation. Each model can be paired with any of the three temporal encoders described in Appendix A.2.

Table 1: Comparison of CaReTS1–4 models

| Model | Arch. | Trend | Deviation | Fusion | Loss |
|-------|-------|-------|-----------|--------|------|
| **CaReTS1** | (a) | Binary label $\hat{d}^{(k)} \in \{+1, -1\}$ | Non-negative deviation $\hat{\delta}^{(k)}$ | $\hat{y}^{(k)} = x_n + \hat{d}^{(k)} \cdot \hat{\delta}^{(k)}$ | $L_{(a)} = L_{\text{ca}} + L_{\text{de}} + L_{\text{op}}$ Eq. (9), (10), (11) |
| **CaReTS2** | (a) | Binary label $\hat{d}^{(k)} \in \{+1, -1\}$ | Non-negative deviations $(\hat{\delta}_{\text{up}}^{(k)}, \hat{\delta}_{\text{down}}^{(k)})$ | If up: $\hat{y}^{(k)} = x_n + \hat{\delta}_{\text{up}}^{(k)}$, else: $\hat{y}^{(k)} = x_n - \hat{\delta}_{\text{down}}^{(k)}$ | $L_{(a)} = L_{\text{ca}} + L_{\text{de}} + L_{\text{op}}$ Eq. (9), (13), (11) |
| **CaReTS3** | (a) | Probabilities $(p_{\text{up}}^{(k)}, p_{\text{down}}^{(k)})$ | Non-negative deviations $(\hat{\delta}_{\text{up}}^{(k)}, \hat{\delta}_{\text{down}}^{(k)})$ | $\hat{y} = x_n + p_{\text{up}}^{(k)} \hat{\delta}_{\text{up}}^{(k)} - p_{\text{down}}^{(k)} \hat{\delta}_{\text{down}}^{(k)}$ | $L_{(a)} = L_{\text{ca}} + L_{\text{de}} + L_{\text{op}}$ Eq. (16), (13), (11) |
| **CaReTS4** | (b) | Probabilities $p^{(k)}$ | Signed deviation $\hat{\delta}^{(k)}$ | $\hat{y} = x_n + \hat{\delta}^{(k)}$ | $L_{(b)} = L_{\text{ca}} + L_{\text{op}}$ Eq. (18), (19) |

### 3.1 CaReTS1

This variant follows architecture (a) with two parallel fully connected (FC) streams. For each forecast step $k$, the trend branch predicts a binary class label $\hat{d}^{(k)} \in \{+1, -1\}$, where $+1$ denotes an upward trend and $-1$ a downward trend. In implementation, a single logit $z^{(k)}$ is predicted and transformed into a probability $p^{(k)} \in (0, 1)$ via the sigmoid function:

$$\hat{d}^{(k)} = \begin{cases} +1, & \text{if } p^{(k)} \geq 0.5, \\ -1, & \text{if } p^{(k)} < 0.5, \end{cases} \tag{6}$$

where

$$p^{(k)} = \frac{1}{1 + e^{-z^{(k)}}}. \tag{7}$$

Meanwhile, the deviation branch predicts a single non-negative $\hat{\delta}^{(k)} \geq 0$, representing the absolute magnitude of change from the latest observation $x_n$, independent of direction.

Finally, the forecast $\hat{y}^{(k)}$ is obtained by combining the predicted trend direction and magnitude:

$$\hat{y}^{(k)} = x_n + \hat{d}^{(k)} \hat{\delta}^{(k)}. \tag{8}$$

The detailed loss of classification, deviation regression, and output prediction are defined as:

$$L_{\mathrm{ca}} = \frac{1}{K} \sum_{k=1}^{K} \mathrm{BCE}\left(p^{(k)}, t^{(k)}\right), \tag{9}$$

$$L_{\mathrm{de}} = \frac{1}{K} \sum_{k=1}^{K} \mathrm{MSE}\left(\hat{\delta}^{(k)}, \delta^{(k)}\right), \tag{10}$$

$$L_{\mathrm{op}} = \frac{1}{K} \sum_{k=1}^{K} \mathrm{MSE}\left(\hat{y}^{(k)}, y^{(k)}\right), \tag{11}$$

where $t^{(k)} \in \{0, 1\}$ is the ground-truth trend label (1 for upward trend, 0 for downward trend), $\delta^{(k)} = \left| y^{(k)} - x_n \right|$ is the true absolute deviation, $\mathrm{MSE}(\cdot, \cdot)$ denotes mean squared error, and $\mathrm{BCE}\left(p^{(k)}, t^{(k)}\right) = -\left[t^{(k)} \log p^{(k)} + \left(1 - t^{(k)}\right) \log \left(1 - p^{(k)}\right)\right]$ denotes the binary cross-entropy loss. Note that the ground-truth trend label $t^{(k)}$ is used for BCE loss, which corresponds to $\hat{d}^{(k)} = +1$ when $t^{(k)} = 1$ and $\hat{d}^{(k)} = -1$ when $t^{(k)} = 0$. CaReTS1's simple architecture enables efficient parallel learning of trend movement and deviation magnitude, making the model both tractable and interpretable. Nevertheless, applying a uniform deviation magnitude across either directions in the trend means that any misclassification of trend inevitably results in forecast errors, with no capacity to capture direction-specific variations.

## 3.2 CaReTS2

CaReTS2 also adopts architecture (a) and retains the binary trend classifier from CaReTS1, but addresses one of CaReTS1's limitations: the inability to differentiate magnitude patterns between upward and downward movements. Specifically, CaReTS2 replaces the single deviation output with direction-specific deviations, allowing the model to learn separate regression functions for positive and negative trends. This provides greater flexibility in capturing asymmetric dynamics or time series behaviors. Identical to CaReTS1, CaReTS2 outputs a binary trend label $\hat{d}^{(k)}$ as defined in (6). In contrast, its deviation branch produces two non-negative, direction-specific estimates: $\hat{\delta}\mathrm{up}^{(k)}$ for upward movements and $\hat{\delta}\mathrm{down}^{(k)}$ for downward movements. The final forecast then combines the predicted direction with the corresponding deviation, giving:

$$\hat{y}^{(k)} = \begin{cases} x_n + \hat{\delta}_{\mathrm{up}}^{(k)}, & \text{if } \hat{d}^{(k)} = +1, \\ x_n - \hat{\delta}_{\mathrm{down}}^{(k)}, & \text{if } \hat{d}^{(k)} = -1. \end{cases} \tag{12}$$

The loss function retains the classification term $L_{\mathrm{ca}}$ and output prediction term $L_{\mathrm{op}}$ from CaReTS1. The deviation loss $L_{\mathrm{de}}$, however, is computed using the deviation estimate corresponding to the ground-truth trend direction:

$$L_{\mathrm{de}} = \frac{1}{K} \sum_{k=1}^{K} \left[ t^{(k)} \mathrm{MSE}\left(\hat{\delta}_{\mathrm{up}}^{(k)}, \delta_{\mathrm{up}}^{(k)}\right) + \left(1 - t^{(k)}\right) \mathrm{MSE}\left(\hat{\delta}_{\mathrm{down}}^{(k)}, \delta_{\mathrm{down}}^{(k)}\right) \right], \tag{13}$$

where $\delta_{\mathrm{up}}^{(k)} = \max\left(y^{(k)} - x_n, 0\right)$ and $\delta_{\mathrm{down}}^{(k)} = \max\left(x_n - y^{(k)}, 0\right)$ are the true upward and downward deviations, respectively.

## 3.3 CaReTS3

Similarly, CaReTS3 is based on architecture (a) and adopts the same deviation branch as CaReTS2, producing two separate non-negative estimates: $\hat{\delta}\mathrm{up}^{(k)}$ for upward deviations and $\hat{\delta}\mathrm{down}^{(k)}$ for downward deviations. The key innovation of CaReTS3 lies in the soft probabilistic trend modeling. Instead of producing a hard binary decision, the trend branch first generates a pair of logits $(z_{\mathrm{up}}^{(k)}, z_{\mathrm{down}}^{(k)})$, which are then transformed via a softmax into the output probabilities $(p_{\mathrm{up}}^{(k)}, p_{\mathrm{down}}^{(k)})$:

$$p_{\mathrm{up}}^{(k)} = \frac{e^{z_{\mathrm{up}}^{(k)}}}{e^{z_{\mathrm{up}}^{(k)}} + e^{z_{\mathrm{down}}^{(k)}}}, \quad p_{\mathrm{down}}^{(k)} = 1 - p_{\mathrm{up}}^{(k)}. \tag{14}$$

Unlike selecting a single deviation value based on a hard sign decision, CaReTS3 fuses the two deviation predictions in a soft-weighted manner:

$$\hat{y}^{(k)} = x_n + p_{\text{up}}^{(k)} \hat{\delta}_{\text{up}}^{(k)} - p_{\text{down}}^{(k)} \hat{\delta}_{\text{down}}^{(k)}. \tag{15}$$

This formulation allows both deviation predictions to contribute proportionally to the final forecast, enabling smoother transitions between upward and downward trends and potentially improving robustness when the trend movement is uncertain.

The loss design is same as CaReTS2 in terms of the deviation loss $L_{\text{de}}$ and the output prediction loss $L_{\text{op}}$, but the classification loss $L_{\text{ca}}$ is redefined to handle probabilistic outputs:

$$L_{\text{ca}} = \frac{1}{K} \sum_{k=1}^{K} \text{CE}(\mathbf{p}^{(k)}, \mathbf{t}^{(k)}), \tag{16}$$

where $\mathbf{p}^{(k)} = \big(p_{\text{up}}^{(k)}, p_{\text{down}}^{(k)}\big)$, $\quad \mathbf{t}^{(k)} = \big(t_{\text{up}}^{(k)}, t_{\text{down}}^{(k)}\big)$, with $t_{\text{up}}^{(k)} \in \{0,1\}$, $t_{\text{down}}^{(k)} \in \{0,1\}$ denoting the ground truth vector (1 for upward trend, 0 for downward trend), and the categorical cross-entropy is defined as: $\text{CE}(\mathbf{p}^{(k)}, \mathbf{t}^{(k)}) = -\Big[t_{\text{up}}^{(k)} \log p_{\text{up}}^{(k)} + t_{\text{down}}^{(k)} \log p_{\text{down}}^{(k)}\Big]$.

## 3.4 CaReTS4

CaReTS4 adopts architecture (b) and represents a sequential dual-stream approach, where the trend prediction stage precedes and conditions the deviation estimation stage. For each forecast step $k$, the model outputs the trend probability $p^{(k)}$ using a softmax-based classifier (similar to the trend branch of CaReTS1 and CaReTS2). Then, the predicted trend probabilities are concatenated with the temporal feature vector $\mathbf{h} \in \mathbb{R}^d$ extracted by the encoder, i.e., $\mathbf{h}' = \big[\mathbf{h}, \mathbf{p}\big]$. This operation allows the subsequent regression branch to condition its deviation estimation on the predicted trend context. Using the fused feature vector $\mathbf{h}'$ as input, the model predicts a single signed deviation $\hat{\boldsymbol{\delta}}$, which may be positive or negative. This design differs fundamentally from CaReTS1–CaReTS3, where deviations were constrained to be non-negative and combined with a separate trend sign.

The final forecast is obtained as:
$$\hat{y}^{(k)} = x_n + \hat{\delta}^{(k)}. \tag{17}$$

In contrast to earlier variants, CaReTS4 does not include a separate deviation loss $L_{\text{de}}$. Instead, the training jointly optimizes two objectives:

$$L_{\text{ca}} = \frac{1}{K} \sum_{k=1}^{K} \text{CE}\big(p^{(k)}, t^{(k)}\big), \tag{18}$$

$$L_{\text{op}} = \frac{1}{K} \sum_{k=1}^{K} \text{MSE}\big(\hat{y}^{(k)}, y^{(k)}\big), \tag{19}$$

## 4 Experimentation and Evaluation

We conducted a comprehensive evaluation of CaReTS1–4 on two distinct time-series forecasting tasks (Yao et al., 2025b;a): (i) electricity price forecasting and (ii) import/export power demand (i.e., unmet power) forecasting , both spanning one year with 8,784 hourly observations. Following the original setup, both tasks adopted a 15-to-6 prediction scheme, where the inputs consist of the month, weekday, and hour of the current time step along with the previous 12 observations of the target variable, and the outputs are the predicted values of the target variable in next 6 time steps. Illustrations of the two time series are provided in Appendix A.3, and detailed dataset descriptions can be found in Sec. 5.2 of Yao et al. (2025b). All model evaluations were performed using 10-fold cross-validation (CV), with the mean and standard deviation reported. Implementation details are provided in Appendix A.4. The experiments proceeded in three stages: we first evaluated the effectiveness of the proposed CaReTS architecture against three designed baselines (structural details and motivations in Appendix A.5), then assessed the superiority of multi-task learning, and finally compared CaReTS with 10 state-of-the-art (SOTA) forecasting algorithms.

## 4.1 CaReTS Architecture Evaluation

We first evaluated CaReST 1–4 with three encoders (i.e., CNN, LSTM, and Transformer) on electricity price and unmet power time series. Table 2 reports the average RMSE for multi-step forecasting (mean ± std on 10-fold CV), while Figures 2 and 3 present the corresponding RMSE results on the test set. It can be observed that the proposed CaReST 2–4 models outperformed all baselines on both variables in the test set, regardless of the encoder employed. Among these, CaReST2 achieved the best overall performance, yielding the lowest RMSE in four cases and the second-best results in two others (as marked with '★' in the two figures). Within its configurations, CaReST2 combined with the Transformer encoder represented the best-performing setup, achieving the lowest RMSE of $0.0691 \pm 0.0018$ for unmet power and $0.0465 \pm 0.0012$ for electricity price. However, CaReST1 did not demonstrate a clear advantage over the baselines, which can be attributed to the simplified design of its deviation branch.

Table 2: Average RMSE (mean ± std) for multi-step forecasting across approaches (test set)

| Approach | Unmet power | | | Electricity price | | |
|---|---|---|---|---|---|---|
| | Train | Validation | Test | Train | Validation | Test |
| **LSTM** | | | | | | |
| Baseline1 | $0.0460 \pm 0.0040$ | $0.0666 \pm 0.0030$ | $0.0758 \pm 0.0016$ | $0.0198 \pm 0.0017$ | $0.0378 \pm 0.0027$ | $0.0533 \pm 0.0011$ |
| Baseline2 | $0.0454 \pm 0.0046$ | $0.0691 \pm 0.0016$ | $0.0754 \pm 0.0016$ | $0.0215 \pm 0.0019$ | $0.0423 \pm 0.0018$ | $0.0536 \pm 0.0017$ |
| Baseline3 | $0.0453 \pm 0.0049$ | $0.0682 \pm 0.0021$ | $0.0761 \pm 0.0027$ | $0.0218 \pm 0.0015$ | $0.0393 \pm 0.0013$ | $0.0500 \pm 0.0022$ |
| CaReTS1 | $0.0550 \pm 0.0033$ | $0.0723 \pm 0.0021$ | $0.0767 \pm 0.0016$ | $0.0265 \pm 0.0020$ | $0.0427 \pm 0.0017$ | $0.0488 \pm 0.0011$ |
| CaReTS2 | $0.0512 \pm 0.0023$ | $0.0684 \pm 0.0025$ | $\mathbf{0.0744 \pm 0.0010}$ | $0.0262 \pm 0.0019$ | $0.0400 \pm 0.0025$ | $0.0486 \pm 0.0013$ |
| CaReTS3 | $0.0474 \pm 0.0036$ | $0.0691 \pm 0.0030$ | $0.0750 \pm 0.0021$ | $0.0240 \pm 0.0018$ | $0.0408 \pm 0.0020$ | $0.0491 \pm 0.0013$ |
| CaReTS4 | $0.0519 \pm 0.0032$ | $0.0714 \pm 0.0019$ | $0.0755 \pm 0.0021$ | $0.0314 \pm 0.0014$ | $0.0435 \pm 0.0024$ | $\mathbf{0.0481 \pm 0.0015}$ |
| **CNN** | | | | | | |
| Baseline1 | $0.0619 \pm 0.0024$ | $0.0679 \pm 0.0022$ | $0.0711 \pm 0.0008$ | $0.0410 \pm 0.0015$ | $0.0463 \pm 0.0018$ | $0.0505 \pm 0.0011$ |
| Baseline2 | $0.0550 \pm 0.0026$ | $0.0661 \pm 0.0021$ | $0.0731 \pm 0.0020$ | $0.0317 \pm 0.0015$ | $0.0408 \pm 0.0017$ | $0.0489 \pm 0.0010$ |
| Baseline3 | $0.0576 \pm 0.0020$ | $0.0655 \pm 0.0020$ | $0.0704 \pm 0.0012$ | $0.0310 \pm 0.0019$ | $0.0413 \pm 0.0018$ | $0.0490 \pm 0.0011$ |
| CaReTS1 | $0.0696 \pm 0.0012$ | $0.0739 \pm 0.0029$ | $0.0738 \pm 0.0014$ | $0.0443 \pm 0.0013$ | $0.0499 \pm 0.0013$ | $0.0497 \pm 0.0009$ |
| CaReTS2 | $0.0658 \pm 0.0013$ | $0.0694 \pm 0.0025$ | $0.0695 \pm 0.0013$ | $0.0427 \pm 0.0010$ | $0.0470 \pm 0.0017$ | $\mathbf{0.0473 \pm 0.0007}$ |
| CaReTS3 | $0.0609 \pm 0.0020$ | $0.0665 \pm 0.0022$ | $\mathbf{0.0692 \pm 0.0010}$ | $0.0377 \pm 0.0013$ | $0.0443 \pm 0.0015$ | $0.0474 \pm 0.0008$ |
| CaReTS4 | $0.0626 \pm 0.0018$ | $0.0678 \pm 0.0022$ | $0.0696 \pm 0.0015$ | $0.0428 \pm 0.0014$ | $0.0475 \pm 0.0018$ | $0.0482 \pm 0.0012$ |
| **Transformer** | | | | | | |
| Baseline1 | $0.0561 \pm 0.0084$ | $0.0683 \pm 0.0066$ | $0.0755 \pm 0.0055$ | $0.0322 \pm 0.0025$ | $0.0412 \pm 0.0028$ | $0.0507 \pm 0.0018$ |
| Baseline2 | $0.0530 \pm 0.0056$ | $0.0683 \pm 0.0036$ | $0.0750 \pm 0.0037$ | $0.0359 \pm 0.0037$ | $0.0436 \pm 0.0031$ | $0.0511 \pm 0.0031$ |
| Baseline3 | $0.0542 \pm 0.0044$ | $0.0667 \pm 0.0030$ | $0.0715 \pm 0.0024$ | $0.0353 \pm 0.0037$ | $0.0443 \pm 0.0034$ | $0.0491 \pm 0.0015$ |
| CaReTS1 | $0.0583 \pm 0.0022$ | $0.0702 \pm 0.0028$ | $0.0724 \pm 0.0027$ | $0.0341 \pm 0.0016$ | $0.0444 \pm 0.0026$ | $0.0473 \pm 0.0010$ |
| CaReTS2 | $0.0588 \pm 0.0016$ | $0.0686 \pm 0.0016$ | $\mathbf{0.0691 \pm 0.0018}$ | $0.0333 \pm 0.0020$ | $0.0445 \pm 0.0027$ | $\mathbf{0.0465 \pm 0.0012}$ |
| CaReTS3 | $0.0536 \pm 0.0026$ | $0.0665 \pm 0.0022$ | $0.0699 \pm 0.0019$ | $0.0327 \pm 0.0017$ | $0.0428 \pm 0.0028$ | $0.0487 \pm 0.0009$ |
| CaReTS4 | $0.0588 \pm 0.0031$ | $0.0696 \pm 0.0024$ | $0.0716 \pm 0.0011$ | $0.0375 \pm 0.0027$ | $0.0453 \pm 0.0022$ | $0.0466 \pm 0.0017$ |

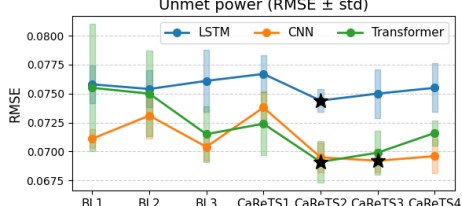

Figure 2: RMSE on power across approaches

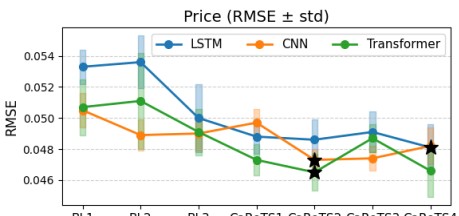

Figure 3: RMSE on price across approaches

Table 3 presents the average trend prediction accuracy achieved by the classification branch in multi-step forecasting, evaluated with CaReTS1-4 using different encoders on both electricity price and unmet power series. For CaReTS3 and CaReTS4, the predicted trend is defined by the direction with the higher probability ($P_{up}$ or $P_{down}$), which is then used to compute the trend prediction accuracy. All variants achieved over 90% accuracy, confirming the framework's ability to capture temporal dynamics. Among the encoders, Transformer consistently outperformed LSTM and CNN, with the CaReTS2-Transformer combination yielding the highest classification accuracy, which aligns with the RMSE results above. Figures 4 and 5, obtained using this best-performing CaReTS2–Transformer model on test set, further illustrate the evolution of classification accuracy and RMSE across six forecasting steps. As expected, RMSE gradually increased with longer forecast horizons due to error accumulation, consistent with prior studies (Yao et al., 2025b; Yunpeng et al., 2017; Venkatraman et al., 2015). Interestingly, trend classification accuracy did not exhibit a

declining pattern, indicating the robustness of the proposed framework in maintaining reliable trend detection even for an extended period of forecasting. This stability arises from the complementary design of the framework: the classification branch captures macro-level trend directions to safeguard long-term consistency, while the regression branch refines micro-level predictions to ensure accuracy.

Table 3: Average trend accuracy for multi-step forecasting across approaches (test set)

| Encoder | CaReTS1 | CaReTS2 | CaReTS3 | CaReTS4 |
|---|---|---|---|---|
| **Unmet power** | | | | |
| LSTM | $0.9111 \pm 0.0020$ | $0.9096 \pm 0.0019$ | $0.9086 \pm 0.0030$ | $0.9068 \pm 0.0041$ |
| CNN | $0.9125 \pm 0.0032$ | $0.9127 \pm 0.0033$ | $0.9125 \pm 0.0032$ | $0.9140 \pm 0.0020$ |
| Transformer | $0.9191 \pm 0.0032$ | $\mathbf{0.9192 \pm 0.0022}$ | $0.9168 \pm 0.0029$ | $0.9166 \pm 0.0025$ |
| **Electricity price** | | | | |
| LSTM | $0.9073 \pm 0.0027$ | $0.9071 \pm 0.0030$ | $0.9066 \pm 0.0032$ | $0.9056 \pm 0.0038$ |
| CNN | $0.9032 \pm 0.0015$ | $0.9036 \pm 0.0030$ | $0.9024 \pm 0.0041$ | $0.9016 \pm 0.0043$ |
| Transformer | $0.9142 \pm 0.0029$ | $\mathbf{0.9146 \pm 0.0019}$ | $0.9135 \pm 0.0021$ | $0.9136 \pm 0.0051$ |

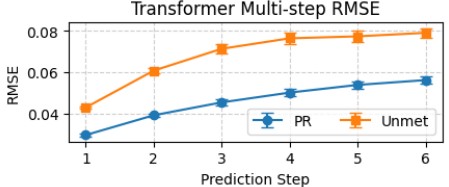

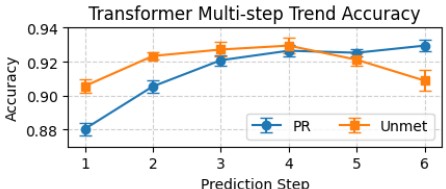

Figure 4: RMSE across forecasting steps using CaReTS2-Transformer

Figure 5: Trend accuracy across forecasting steps using CaReTS2-Transformer

## 4.2 MULTI-TASK LEARNING EVALUATION

We then took the Transformer encoder as a representative case to further evaluate the effectiveness of the multi-task learning mechanism in CaReTS1-4. Table 4 shows the comparison between multi-task and single-task learning. Here, the single-task setting used the same backbone network but with only the output prediction loss ($L_{op}$) optimized, while disregarding the classification ($L_{ca}$) and deviation ($L_{de}$) losses. To ensure the classification branch remained trainable under this setting, the trend direction was implemented in a continuous form, allowing gradients to propagate through the stream. The time used was reported as the average per fold across 10-fold cross-validation. It can be observed that multi-task learning achieved lower RMSE, suggesting that joint optimization promotes complementary learning rather than task interference. By explicitly separating trend classification and deviation estimation within the CaReTS architecture, the model provides more transparent insights into decision factors. Under multi-task learning, the trend classification branch attained over 91% accuracy, yielding reliable trend predictions. Conversely, single-task training, which lacks explicit classification supervision, achieved lower accuracy. Regarding computational cost, the overhead of multi-task learning is negligible. The additional parameters are limited to three task-weight scalars, leaving the overall model size virtually unchanged. Moreover, forward computation introduces only a few extra exponential operations, and backward propagation involves calculation of gradients of only these three scalars, resulting in no significant increase in runtime compared with single-task training.

## 4.3 COMPARISON WITH SOTA ALGORITHMS

Table 5 summarizes the results of ten representative state-of-the-art (SOTA) algorithms, which are compared against the proposed multi-task CaReTS variants in Table 4. The results are reported under the 15-input–6-output setting, while comparisons and further analysis with other input–output configurations are provided in Appendix A.6. The comparison clearly demonstrates that CaReTS achieves state-of-the-art performance, particularly in reducing RMSE while maintaining strong trend consistency. For unmet power forecasting, CaReTS2 and CaReTS3 yielded the lowest RMSE values (0.0691 and 0.0699, respectively) with trend accuracy above 0.916, outperforming the best SOTA model TimeXer (RMSE = 0.0700, trend accuracy = 0.9066). Even CaReTS1 and CaReTS4 delivered competitive results, with their performance closely following that of TimeXer. For electricity

Table 4: Test results of CaReTS1–4 with Transformer: multi-task vs. single-task learning

| Approach | Proposed multi-task | | | Single-task | | |
|---|---|---|---|---|---|---|
| | RMSE | Trend Acc. | Time (s) | RMSE | Trend Acc. | Time (s) |
| | | | Unmet power | | | |
| CaReTS1 | $0.0724 \pm 0.0027$ | $0.9191 \pm 0.0032$ | 253.39 | $0.0758 \pm 0.0036$ | $0.8874 \pm 0.0046$ | 216.37 |
| CaReTS2 | $\mathbf{0.0691 \pm 0.0018}$ | $\mathbf{0.9192 \pm 0.0022}$ | 256.69 | $0.0704 \pm 0.0029$ | $0.9060 \pm 0.0023$ | 261.37 |
| CaReTS3 | $0.0699 \pm 0.0019$ | $0.9168 \pm 0.0029$ | 296.31 | $0.0721 \pm 0.0017$ | $0.8965 \pm 0.0036$ | 236.11 |
| CaReTS4 | $0.0716 \pm 0.0011$ | $0.9166 \pm 0.0025$ | 306.44 | $0.0716 \pm 0.0026$ | $0.9053 \pm 0.0058$ | 313.37 |
| | | | Electricity price | | | |
| CaReTS1 | $0.0473 \pm 0.0010$ | $0.9142 \pm 0.0029$ | 357.20 | $0.0539 \pm 0.0023$ | $0.8663 \pm 0.0028$ | 333.96 |
| CaReTS2 | $\mathbf{0.0465 \pm 0.0012}$ | $\mathbf{0.9146 \pm 0.0019}$ | 388.49 | $0.0470 \pm 0.0020$ | $0.8939 \pm 0.0033$ | 379.79 |
| CaReTS3 | $0.0487 \pm 0.0009$ | $0.9135 \pm 0.0021$ | 401.18 | $0.0474 \pm 0.0012$ | $0.8860 \pm 0.0014$ | 386.43 |
| CaReTS4 | $0.0466 \pm 0.0017$ | $0.9136 \pm 0.0051$ | 321.93 | $0.0472 \pm 0.0018$ | $0.8889 \pm 0.0042$ | 318.14 |

price forecasting, the proposed CaReTS family (CaReTS1–4) demonstrated consistently strong performance, outperforming all SOTA algorithms except TimeXer. Among them, CaReTS2 achieved the best balance, delivering a competitive RMSE (0.0465) together with the highest trend accuracy (0.9146). While TimeXer obtained the lowest RMSE (0.0463), it suffered from lower trend accuracy (0.9013) and required considerably more computation effort. A distinctive advantage of CaReTS lies in the consistently high accuracy in the trend prediction across all variants. This improvement stems from our multi-task optimization design, which explicitly separates the trend classification, enhancing the learning efficiency and results interpretability.

From the perspective of efficiency, CaReTS runs within a moderate cost ($\approx$200–400s), which is much faster than heavier architectures such as Autoformer (>460s) or SOIT2FNN-MO (>860s). Although slower than lightweight baselines (e.g., Nlinear/Dlinear <70s and TimeMixer < 85s), CaReTS achieves a favorable trade-off, where the additional computation is modest compared to the substantial accuracy gains.

Table 5: Test results of SOTA algorithms on unmet power and electricity price forecasting

| Approach | Unmet power | | | Electricity price | | |
|---|---|---|---|---|---|---|
| | RMSE | Trend Acc. | Time (s) | RMSE | Trend Acc. | Time (s) |
| Autoformer (Wu et al., 2021) | $0.0731 \pm 0.0009$ | $0.8891 \pm 0.0036$ | 510.05 | $0.0487 \pm 0.0021$ | $0.8713 \pm 0.0073$ | 467.97 |
| FEDformer (Zhou et al., 2022) | $0.0908 \pm 0.0005$ | $0.8345 \pm 0.0023$ | 222.80 | $0.0874 \pm 0.0011$ | $0.7477 \pm 0.0086$ | 239.34 |
| Non-stationary Transformer (Liu et al., 2022) | $0.1588 \pm 0.0025$ | $0.7384 \pm 0.0076$ | 541.35 | $0.1176 \pm 0.0036$ | $0.6970 \pm 0.0172$ | 422.41 |
| D-CNN-LSTM(Yao et al., 2022) | $0.0732 \pm 0.0009$ | $0.8924 \pm 0.0034$ | 103.28 | $0.0573 \pm 0.0012$ | $0.8821 \pm 0.0067$ | 112.02 |
| TimesNet (Wu et al., 2023) | $0.0729 \pm 0.0012$ | $0.8990 \pm 0.0028$ | 273.15 | $0.0500 \pm 0.0015$ | $0.8737 \pm 0.0074$ | 314.40 |
| Dlinear (Zeng et al., 2023) | $0.0859 \pm 0.0004$ | $0.8335 \pm 0.0028$ | 68.75 | $0.0701 \pm 0.0005$ | $0.7337 \pm 0.0085$ | 70.07 |
| Nlinear (Zeng et al., 2023) | $0.1327 \pm 0.0002$ | $0.8033 \pm 0.0017$ | 48.44 | $0.1060 \pm 0.0004$ | $0.7259 \pm 0.0036$ | 50.33 |
| TimeXer (Wang et al., 2024c) | $0.0700 \pm 0.0022$ | $0.9066 \pm 0.0022$ | 448.62 | $0.0463 \pm 0.0013$ | $0.9013 \pm 0.0054$ | 573.75 |
| TimeMixer (Wang et al., 2024a) | $0.1471 \pm 0.0008$ | $0.6983 \pm 0.0048$ | 76.25 | $0.1134 \pm 0.0010$ | $0.5831 \pm 0.0100$ | 84.60 |
| SOIT2FNN-MO (Yao et al., 2025b) | $0.1638 \pm 0.0012$ | $0.7021 \pm 0.0020$ | 863.05 | $0.1439 \pm 0.0018$ | $0.7153 \pm 0.0042$ | 926.81 |

Despite these encouraging results, our evaluation remains limited for long-horizon forecasting constrained by the available GPU resources. Nevertheless, the proposed approach shows strong potential for reliable multi-step time series prediction. As illustrated in Figure 5, the trend classification accuracy remains stable (even shows a slight improvement) as the prediction horizon increases, rather than deteriorating. This would provide an insightful indicator that supports the applicability of CaReTS algorithms to situations requiring extended periods of forecasting. Here, we welcome future investigations, particularly by research groups with greater computational capacity, to further validate and extend the CaReTS framework in large-scale and long-horizon forecasting scenarios.

## 5 CONCLUSION

We proposed CaReTS, a dual-stream multi-task framework, for multi-step time series forecasting that separates trend classification from deviation estimation. An uncertainty-aware weighting was employed to enable multi-task optimization. Four variants (CaReTS1–4) based on this framework were designed to support various temporal encoders, with Transformer-based CaReTS2 achieving the best performance. Experiments showed that CaReTS outperformed SOTA algorithms in both value forecasting and trend classification, while the dual-stream design improves the explainability of prediction with manageable compute resource.

## REPRODUCIBILITY STATEMENT

Anonymous code is available at: https://anonymous.4open.science/r/CaReTS-6A8F/README.md

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

# A  APPENDIX

## A.1  RELATED WORK

Deep learning has substantially advanced time series forecasting, with models ranging from CNNs and RNNs to Transformers. CNN-based methods (Durairaj & Mohan, 2022; Sadouk, 2019) are effective in extracting local spatial or temporal patterns, while RNN variants such as LSTM and GRU (Elsworth & Güttel, 2020; Zhang et al., 2024) are good at capturing sequential dynamics over time. Hybrid models such as D-CNN-LSTM (Yao et al., 2022) combine convolution and recurrence to capture both local and sequential structures. At present, Transformer-based architectures (Vaswani et al., 2017) have emerged as the dominant backbone in time series forecasting. Representative examples include Autoformer (Wu et al., 2021), which incorporates trend-seasonal decomposition and autocorrelation mechanisms; FEDformer (Zhou et al., 2022), which introduces frequency-domain decomposition; TimesNet (Wu et al., 2023), which captures temporal variations in a 2D representation; and Non-stationary Transformer (Liu et al., 2022), which addresses distributional shifts in nonstationary series. In parallel, researchers have also explored alternatives beyond pure Transformer architectures. For instance, TimeMixer (Wang et al., 2024a) introduces a lightweight MLP-based design for multiscale temporal mixing, while TimeXer (Wang et al., 2024c) focuses on jointly modeling endogenous and exogenous signals through tailored interaction mechanisms. While highly effective, these models predominantly adopt a regression-only learning objective, limiting interpretability in multi-step prediction.

A complementary line of research focuses on decomposition and interpretability. DLinear and NLinear (Zeng et al., 2023) demonstrate that simple linear trend–seasonal decomposition can outperform complex architectures. Transformer variants including ETSformer (Woo et al., 2022), Autoformer (Wu et al., 2021), and FEDformer (Zhou et al., 2022) explicitly model trend, seasonal, or frequency components, thereby enhancing interpretability. However, these methods largely operate at the input or representation level, without directly disentangling the prediction targets. In contrast, CaReTS introduces an output-level decomposition, separating macro-level trends via classification from micro-level fluctuations via regression, which improves both predictive accuracy and interpretability.

Another emerging direction explores multi-task and modular architectures to capture heterogeneous temporal dynamics. TimeXer explicitly distinguishes endogenous from exogenous signals, while TimesNet leverages multi-period modules to model diverse temporal scales. Attention-free MLP-based designs such as TimeMixer (Wang et al., 2024a), LightTS (Zhang et al., 2022), and TSMixer (Chen et al., 2023) achieve competitive results with lightweight architectures. These approaches primarily focus on input-level modularization. By contrast, CaReTS introduces an output-level dual-stream framework that jointly optimizes classification and regression objectives under an uncertainty-aware loss, providing a new perspective on multi-task learning for time series forecasting.

In summary, prior studies have advanced the field in three main areas: encoder innovations (e.g., Informer and TimesNet), signal-level decompositions (e.g., Autoformer, FEDformer, DLinear, and NLinear), and lightweight or multi-branch architectures (e.g., TimeMixer and TimeXer). CaReTS follows the decomposition principle, but explicitly disentangling trend directions from deviation magnitudes. This dual-stream design complements existing encoder improvements while enhancing interpretability, and the proposed uncertainty-aware multi-task loss introduces adaptive task balancing - an aspect rarely explored in current forecasting frameworks.

## A.2 Temporal Encoder

Three typical temporal encoding algorithms, Convolutional Neural Networks (CNNs), Long Short-Term Memory networks (LSTMs), and Transformers, are considered to extract sequential features from the input time series. The structures of these encoders are illustrated in Figure 2.

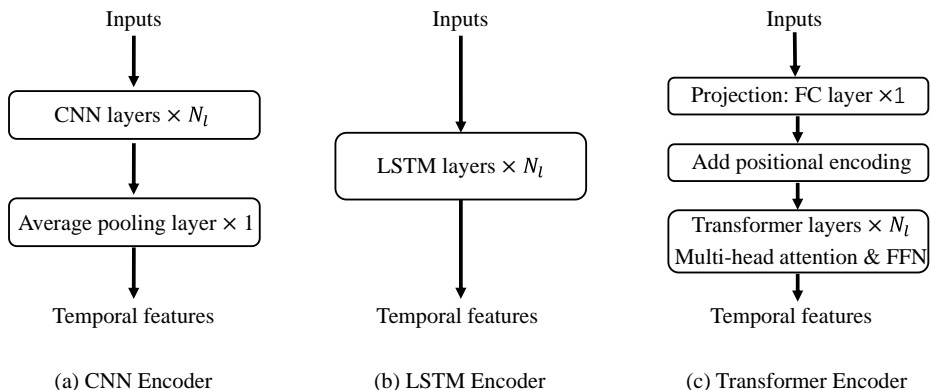

      (a) CNN Encoder        (b) LSTM Encoder        (c) Transformer Encoder

Figure A.1: Three typical temporal encoders

**CNN Encoder:** The CNN-based encoder consists of $N_l$ stacked convolutional layers, designed to capture local temporal patterns and dependencies in the input sequence. The convolutional output is subsequently aggregated through a single average pooling layer to produce compact temporal feature representations.

**LSTM Encoder:** The LSTM-based encoder is composed of $N_l$ stacked LSTM layers, capable of modelling long-range dependencies in sequential data. By processing the input sequence step-by-step, the LSTM encoder generates hidden state sequences enriched with historical contextual information, thereby extracting global temporal dynamics.

**Transformer Encoder:** The Transformer-based encoder begins with a fully connected projection layer that maps the input to a unified feature dimension, followed by the addition of positional encodings to retain sequential order information. The transformed inputs are then processed by $N_l$ standard Transformer encoder layers, each consisting of multi-head self-attention and feed-forward network modules, enabling the capture of complex and global temporal dependencies.

Additional implementation details regarding the three encoders are provided in Appendix A.4.

## A.3 Visualization of Two Time-Series Datasets

The training set contains 6,048 points and the test set contains 2,736 points, for both unmet power and electricity price. As illustrated in Figures A.2 and A.3, the two time series exhibit markedly different patterns to evaluate the robustness and generalization ability of forecasting algorithms under diverse conditions.

## A.4 Implementation Details

**Configurations:** Experiments were implemented in Python and executed on Google Colab with a single T4 GPU. CaReTS used two fully connected layers with 64 hidden units for both the trend classification branch and the deviation estimation branch. Training was performed for up to 600 epochs with early stopping if no improvement is observed for 50 consecutive epochs. The Adam optimizer was employed with a learning rate of 0.001, and the batch size was set to 64. The random seed is fixed at 2025. All datasets were preprocessed using Min-Max normalization, and ReLU was applied as the activation function. For the encoder design, $N_l = 2$ layers with 64 hidden units were adopted in three encoder variants. Specifically, a kernel size of 3 with padding of 1 was used

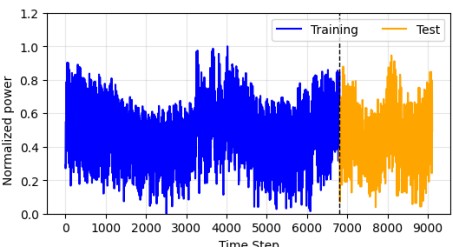 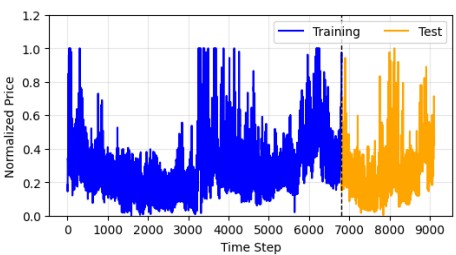

Figure A.2: Illustration of unmet power      Figure A.3: Illustration of electricity price

in the CNN encoder, while all Transformer encoders were configured with 4 attention heads. For the Baseline1–3 algorithms described in Appendix A.5, we set $N_b = N_l = 2$ with fully connected layers of 64 units each.

**10-Fold Cross-Validation:** We adopted 10-fold cross-validation to robustly evaluate our model. The training set was partitioned into 10 equally sized folds; for each fold, the model was trained on 9 folds and validated on the remaining fold, producing independent performance metrics. After completing all 10 folds, we reported the mean and standard deviation of the metrics on the held-out test sets. Each fold was treated equally, and the procedure was fully isolated from the final test set to prevent any data leakage.

## A.5 THREE DESIGNED BASELINES

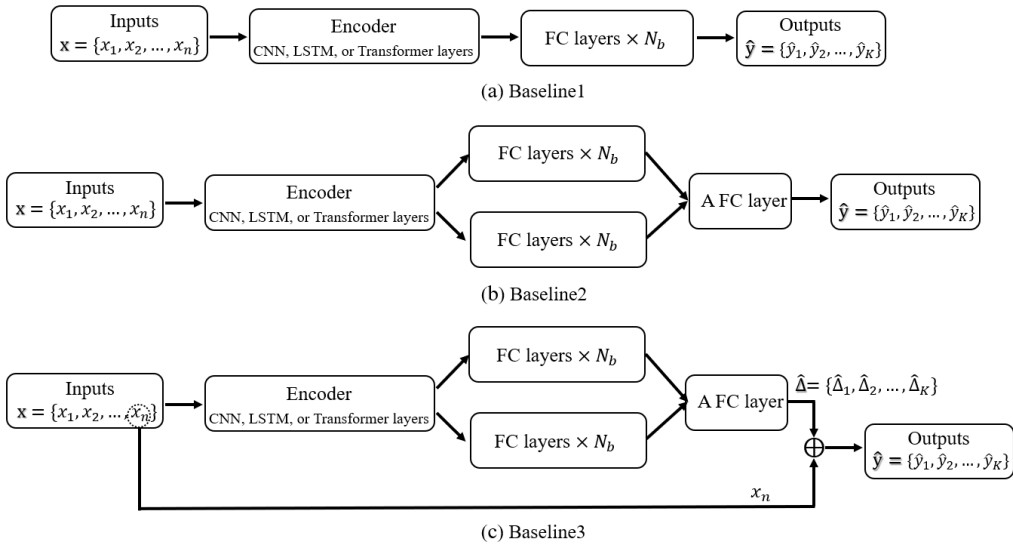

Figure A.4: Structures of three new baselines

Here, we design three baselines to provide fair and transparent comparisons, as illustrated in Figure A.4. Baseline3 adopted a structure closely aligned with our proposed CaReTS, but replaced the fusion formulation in CaReTS with a single fully connected layer. Baseline2 simplified Baseline3 by removing the residual connection, such that the network directly outputs $\hat{y}$ instead of predicting the deviation with respect to the latest observation $x_n$. Baseline1 corresponds to a more conventional encoder-decoder design, where the encoder (CNN, LSTM, or Transformer layers) is directly followed by $N_b$ fully connected layers that map the input sequence to multi-step predictions. In summary, these baselines were constructed to progressively reduce modeling capacity, thereby en-

abling us to clearly demonstrate the contribution of each design component and to highlight the effectiveness of the proposed CaReTS framework.

## A.6 Extended Experimental Results for Alternative Input–Output Settings

Table 6 reports the extended experimental results on the unmet power and electricity price datasets under the 15-4 and 15-8 forecasting settings, where the input length was fixed at 15 and the prediction horizon was either shortened or extended. For each case, the lowest three RMSEs and the highest three trend accuracies among all algorithms are highlighted in bold. It can be observed that our proposed CaReTS1–4 models consistently deliver strong performance, surpassing all other baselines except TimeXer. In particular, CaReTS2 achieved the best overall results, always among the top three in terms of both RMSE and trend accuracy.

Table 6: Comparison of SOTA algorithms for 15-4 and 15-8 multi-step forecasting of unmet power and electricity price

| Approach | 15-4 Unmet power | | 15-4 Electricity price | | 15-8 Unmet power | | 15-8 Electricity price | |
|---|---|---|---|---|---|---|---|---|
| | RMSE | Trend Acc. | RMSE | Trend Acc. | RMSE | Trend Acc. | RMSE | Trend Acc. |
| CaReTS1 | $0.0650 \pm 0.0014$ | $\mathbf{0.9193 \pm 0.0024}$ | $0.0418 \pm 0.0011$ | $\mathbf{0.9091 \pm 0.0022}$ | $\mathbf{0.0763 \pm 0.0021}$ | $\mathbf{0.9095 \pm 0.0030}$ | $0.0521 \pm 0.0024$ | $0.9090 \pm 0.0019$ |
| CaReTS2 | $\mathbf{0.0646 \pm 0.0016}$ | $\mathbf{0.9208 \pm 0.0026}$ | $\mathbf{0.0408 \pm 0.0013}$ | $\mathbf{0.9086 \pm 0.0021}$ | $0.0758 \pm 0.0025$ | $\mathbf{0.9085 \pm 0.0040}$ | $\mathbf{0.0512 \pm 0.0019}$ | $\mathbf{0.9183 \pm 0.0033}$ |
| CaReTS3 | $\mathbf{0.0641 \pm 0.0021}$ | $\mathbf{0.9207 \pm 0.0031}$ | $0.0422 \pm 0.0013$ | $\mathbf{0.9096 \pm 0.0022}$ | $0.0764 \pm 0.0024$ | $0.9066 \pm 0.0025$ | $\mathbf{0.0520 \pm 0.0015}$ | $\mathbf{0.9107 \pm 0.0023}$ |
| CaReTS4 | $0.0654 \pm 0.0023$ | $0.9186 \pm 0.0033$ | $\mathbf{0.0412 \pm 0.0013}$ | $0.9060 \pm 0.0037$ | $\mathbf{0.0756 \pm 0.0029}$ | $\mathbf{0.9090 \pm 0.0050}$ | $\mathbf{0.0518 \pm 0.0013}$ | $\mathbf{0.9185 \pm 0.0011}$ |
| Autoformer | $0.0683 \pm 0.0020$ | $0.8875 \pm 0.0055$ | $0.0437 \pm 0.0018$ | $0.8815 \pm 0.0086$ | $0.0785 \pm 0.0019$ | $0.8856 \pm 0.0031$ | $0.0579 \pm 0.0026$ | $0.8608 \pm 0.0082$ |
| FEDformer | $0.0841 \pm 0.0003$ | $0.8407 \pm 0.0038$ | $0.0843 \pm 0.0006$ | $0.7536 \pm 0.0072$ | $0.1097 \pm 0.0004$ | $0.8165 \pm 0.0028$ | $0.1035 \pm 0.0014$ | $0.7607 \pm 0.0071$ |
| Non-stationary | $0.1408 \pm 0.0018$ | $0.7341 \pm 0.0126$ | $0.1032 \pm 0.0023$ | $0.6842 \pm 0.0157$ | $0.1581 \pm 0.0018$ | $0.7250 \pm 0.0031$ | $0.1286 \pm 0.0015$ | $0.7118 \pm 0.0099$ |
| D-CNN-LSTM | $0.0651 \pm 0.0018$ | $0.8813 \pm 0.0040$ | $0.0503 \pm 0.0015$ | $0.7899 \pm 0.0036$ | $0.0790 \pm 0.0024$ | $0.8663 \pm 0.0027$ | $0.0631 \pm 0.0011$ | $0.8007 \pm 0.0026$ |
| TimesNet | $0.0648 \pm 0.0014$ | $0.8962 \pm 0.0067$ | $0.0434 \pm 0.0021$ | $0.8834 \pm 0.0093$ | $0.0776 \pm 0.0011$ | $0.8939 \pm 0.0017$ | $0.0578 \pm 0.0038$ | $0.8827 \pm 0.0101$ |
| Dlinear | $0.0744 \pm 0.0004$ | $0.8391 \pm 0.0039$ | $0.0664 \pm 0.0006$ | $0.7336 \pm 0.0170$ | $0.0805 \pm 0.0003$ | $0.8179 \pm 0.0015$ | $0.0633 \pm 0.0004$ | $0.7492 \pm 0.0038$ |
| Nlinear | $0.1105 \pm 0.0002$ | $0.8116 \pm 0.0029$ | $0.0903 \pm 0.0005$ | $0.7272 \pm 0.0111$ | $0.1360 \pm 0.0002$ | $0.7855 \pm 0.0017$ | $0.1127 \pm 0.0006$ | $0.7424 \pm 0.0025$ |
| TimeXer | $\mathbf{0.0637 \pm 0.0016}$ | $0.9066 \pm 0.0034$ | $\mathbf{0.0417 \pm 0.0015}$ | $0.8993 \pm 0.0064$ | $0.0769 \pm 0.0033$ | $0.8934 \pm 0.0060$ | $0.0532 \pm 0.0034$ | $0.9049 \pm 0.0042$ |
| TimeMixer | $0.1248 \pm 0.0005$ | $0.6646 \pm 0.0037$ | $0.0971 \pm 0.0014$ | $0.5687 \pm 0.0091$ | $0.1307 \pm 0.0003$ | $0.7125 \pm 0.0011$ | $0.1240 \pm 0.0003$ | $0.6252 \pm 0.0051$ |
| SOIT2FNN-MO | $0.1519 \pm 0.0020$ | $0.6955 \pm 0.0027$ | $0.1287 \pm 0.0022$ | $0.5946 \pm 0.0039$ | $0.1689 \pm 0.0026$ | $0.6886 \pm 0.0023$ | $0.1304 \pm 0.0019$ | $0.6599 \pm 0.0039$ |

