# OpenReview forum: "CaReTS: A Multi-Task Framework Unifying Classification and Regression for Time Series Forecasting"
_ICLR.cc/2026/Conference — Submitted to ICLR 2026_

### Official Review · Reviewer_wCda · 2025-10-30

**Soundness:** 2
**Presentation:** 2
**Contribution:** 1
**Rating:** 2
**Confidence:** 4

**Summary:**

The paper presents a multi-task learning framework by combining classification and regression for multi-step forecasting. The framework was tested with multiple temporal modeling encoders (LSTM, CNN and transformers). Four variants of these models have been proposed and tested with two datasets, Unmet power and electricity price, respectively.

**Strengths:**

1. Paper is clear and well presented.
2. Framework appears to be encoder agnostic and can be easily plugged in any encoder architecture.
3. Modest improvements in the performance for predictions in the two datasets considered.

**Weaknesses:**

The advantages of the proposed architecture are not clear based on the results in the paper. The gains in the performance are very minor compared to traditional encoder architectures to substantiate the multi-task framework.

The models have been tested on two niche datasets. The paper probably needs to consider standard time series foundation model datasets like ETTh1, Weather etc to clearly demonstrate the advantages.

The paper is probably better placed to be in a domain specific conference as the proposed architecture presumably has some advantages in the electricity domain.

**Questions:**

1. Why were standard datasets in time series foundation models not considered for this study? (Ex. TTM paper or Chronos paper datasets)

2. How does the model perform in long term predictions where the trends can be positive and negative? Some details on this component can be included in the paper to make the paper more exhaustive in details.

3. In Table 5, are the times reported only for inferencing?  If yes, that seems really high and a wide spread across the models. Can you please provide more details on how this was computed and how many samples were there for inferencing?

---

> ### Author Response · Authors · 2025-11-27
> **Thanks!**
>
> Thanks for your comments!

---

### Official Review · Reviewer_2mbM · 2025-10-31

**Soundness:** 2
**Presentation:** 3
**Contribution:** 2
**Rating:** 2
**Confidence:** 3

**Summary:**

This paper proposes CaReTS, a multi-task framework for time series forecasting that decomposes predictions into trend classification (directional movement) and deviation regression (magnitude estimation). Four variants (CaReTS1-4) are instantiated using CNN, LSTM, and Transformer encoders, with uncertainty-weighted multi-task losses. Experiments on two hourly energy datasets show competitive RMSE and improved trend accuracy (>91%) compared to ten SOTA methods, including TimeXer and Autoformer. While the approach offers interpretability and computational efficiency, the evaluation scope is limited to small-scale univariate forecasting tasks

**Strengths:**

1. Explicit trend/deviation separation provides actionable insights for energy/finance applications-a genuine practical advantage often overlooked in pure accuracy-focused papers
2. 200-400s training is substantially faster than Autoformer (>460s) and SOIT2FNN-MO (>860s) while maintaining competitive accuracy.
3. Good trend accuracy, remaining stable across forecast horizons (Figure 5) is valuable for risk management applications
4. Comprehensive ablations across encoders, single vs multi-task learning, and architecture variants

**Weaknesses:**

1. Multi-task classification + regression is well-established; the core contribution is applying this to time series with uncertainty weighting. The architectural variants (CaReTS1-4) differ only in fusion; this feels more like hyperparameter exploration than distinct methodological contributions.

2. Narrow evaluation scope:
a. Only 2 datasets (both hourly energy data) from a single domain
b. Missing standard benchmarks (ETTh1/h2, Weather, Traffic, Exchange) used in TimesNet[4]/iTransformer[5]/PatchTST[6].
c. Only the 15-to-6 horizon was evaluated in the main paper; while domain-appropriate for energy trading, comparison with 96/192/336/720-step horizons expected in modern time series papers is absent.
d.Univariate forecasting only. No multivariate output or channel modeling.

3.	Incomplete baseline coverage:
a. Missing recent Transformer architectures: PatchTST (patch-based), iTransformer (channel-inverted), Crossformer (dimension-segment attention).
b. No comparison with interpretable baselines: N-BEATS (decomposition), TFT (temporal fusion), Explainable Boosting Machines.
c. Extended results (Table 6) show similar limitations across 15-4 and 15-8 settings.

4.Shallow analysis:
 a. Why does CaReTS2 outperform CaReTS3/4? The paper lacks an investigation into when probabilistic fusion (CaReTS3) vs hard fusion (CaReTS2) is preferred.
b. No analysis of failure modes or dataset characteristics favors the approach.
c. Uncertainty weighting (Eq. 4-5) benefits are not empirically validated-Table 4 compares multi-task vs single-task, but doesn’t show uncertainty weighting vs fixed.

**Questions:**

1.	Can the framework be extended to multi-output forecasting (e.g., predicting electricity price AND demand simultaneously)? How would the classification branch handle correlated trends across variables?
2.	Can you evaluate on ETTh1/h2 (Weather, Traffic, only if possible), following the {96 or 336} predicts {96,192,336,720} protocol of iTransformer/TimesNet. This enables a fair comparison.
3.	Can you add PatchTST, iTransformer, and at least one interpretable baseline (N-BEATS or TFT)? Given your interpretability focus, comparison with TFT’s variable selection networks is particularly relevant.
4.	Rather than four separate variants, present a single architecture with ablations for: (a) fusion strategy (parallel vs sequential), (b) deviation branches (single vs dual), (c) trend representation (binary vs probabilistic), (d) uncertainty weighting vs fixed weights. Would this be possible to implement?
5.	Can you report FLOPs and memory usage, not just wall-clock time? How does training time scale with sequence length and number of output steps?

---

> ### Author Response · Authors · 2025-11-27
> **Thanks**
>
> Thanks for your comments.

---

### Official Review · Reviewer_rud9 · 2025-10-31

**Soundness:** 2
**Presentation:** 1
**Contribution:** 1
**Rating:** 2
**Confidence:** 2

**Summary:**

•	The paper proposes CaReTS, a multi-task learning framework for time series forecasting that explicitly separates classification and regression into dual-stream architectures. This paper also introduces an uncertainty-aware loss weighting mechanism to balance task contributions during training.

**Strengths:**

•	1. The authors apply the proposed architecture to various architectures, demonstrating its strong adaptability.

•	2. The authors explicitly separate the regression and classification modules, which indeed enhances the model's interpretability.

**Weaknesses:**

•	Jointly optimizing multiple tasks (e.g., regression and classification) through a shared encoder with separate task-specific heads is already a widely adopted approach across various fields[1][2] and does not represent a significant novelty. Additionally, incorporating uncertainty-based loss weighting for adaptive multi-task loss balancing is also widely adopted [3].

•	[1] A Multitask Deep Learning Model for Classification and Regression of Hyperspectral Images: Application to the Large-Scale Dataset

•	[2]Joint Classification and Trajectory Regression of Online Handwriting using a Multi-Task Learning Approach

•	[3] Multi-Task Self-Supervised Time-Series Representation Learning.


•	Considering that 2025a is specifically designed for power grid time-series scenarios, and 2025b also emphasizes the power grid setting in its abstract, the current work, being more general in scope, should follow more generic experimental setups and include evaluations on additional, diverse datasets beyond the power grid domain.


•	The authors should better organize the presentation of their experimental results in tables. Due to the introduction of multiple variants and the architecture evaluation, the current tables appear somewhat cluttered in presenting the information. The authors should place greater emphasis on comparisons with SOTA methods.

**Questions:**

•	The authors need to demonstrate their unique contribution in the design of the multi-task adaptive loss function, as well as justify why they chose uncertainty-based loss over other adaptive loss methods (e.g., Gradient Normalization loss).

---

> ### Author Response · Authors · 2025-11-27
> **Thanks！**
>
> Thanks for your comments.

---

### Meta-Review · Area_Chair_fyua · 2026-01-07

**Summary:**

The paper presents a multi-task learning framework by combining classification and regression for multi-step forecasting. All three reviewers provided negative feedbacks with reasonable concerns, and the authors failed to adequately address them in the rebuttal phase. Therefore, I recommend rejection of this paper. I encourage the authors to revise the paper and resubmit it to future conferences.

**Reviewer Concerns:**

Reviewer rud9 raised concerns on the novelty of the proposed method, experimental setups, and presentation issues.

Reviewer 2mbM agreed with reviewer rud9 on the novelty issue and narrow evaluation setups, and raised additional concerns on the incomplete baseline converage and result analysis.

Reviewer wCda raised concerns on the minor improvements and dataset choices.

Authors can consider the suggestions from reviewers to improve the paper accordingly.

**Reviewer Scores:**

Given that the authors did not provide valid responses to the reviewers' concerns, I think the reviewers' scores will remain the same.

---

### Decision · Program_Chairs · 2026-01-26

Reject